# The Role of Non-Invasive Multimodality Imaging in Chronic Coronary Syndrome: Anatomical and Functional Pathways

**DOI:** 10.3390/diagnostics13122083

**Published:** 2023-06-16

**Authors:** Luca Bergamaschi, Anna Giulia Pavon, Francesco Angeli, Domenico Tuttolomondo, Marta Belmonte, Matteo Armillotta, Angelo Sansonetti, Alberto Foà, Pasquale Paolisso, Andrea Baggiano, Saima Mushtaq, Giulia De Zan, Serena Carriero, Maarten-Jan Cramer, Arco J. Teske, Lysette Broekhuizen, Ivo van der Bilt, Giuseppe Muscogiuri, Sandro Sironi, Laura Anna Leo, Nicola Gaibazzi, Luigi Lovato, Gianluca Pontone, Carmine Pizzi, Marco Guglielmo

**Affiliations:** 1Division of Cardiology, Cardiocentro Ticino Institute, Ente Ospedaliero Cantonale, Via Tesserete, 48, 6900 Lugano, Switzerlandannagiulia.pavon@eoc.ch (A.G.P.); lauraanna.leo@eoc.ch (L.A.L.); 2Cardiology Unit, IRCCS Azienda Ospedaliera-Universitaria di Bologna, 40138 Bologna, Italy; francesco.angeli1992@gmail.com (F.A.); matteo.armillotta2@studio.unibo.it (M.A.); angelo.sansonetti@studio.unibo.it (A.S.); alberto.foa@gmail.com (A.F.); carmine.pizzi@unibo.it (C.P.); 3Department of Medical and Surgical Sciences—DIMEC—Alma Mater Studiorum, University of Bologna, 40138 Bologna, Italy; 4Department of Cardiology, Parma University Hospital, Viale Antonio Gramsci 14, 43126 Parma, Italy; d.tuttolomondo@hotmail.it (D.T.); ngaibazzi@gmail.com (N.G.); 5Cardiovascular Center Aalst, OLV-Clinic, 9300 Aalst, Belgium; belmontemarta1@gmail.com; 6Department of Advanced Biomedical Sciences, University Federico II, 80138 Naples, Italy; pasquale.paolisso@gmail.com; 7Perioperative and Cardiovascular Imaging Department, Centro Cardiologico Monzino IRCCS, 20138 Milan, Italy; andrea.baggiano@cardiologicomonzino.it (A.B.); saima.mushtaq@cardiologicomonzino.it (S.M.); gianluca.pontone@cardiologicomonzino.it (G.P.); 8Department of Clinical Sciences and Community Health, University of Milan, 20122 Milan, Italy; 9Department of Cardiology, University Medical Center Utrecht, 3584 CX Utrecht, The Netherlands; giuliadezan@gmail.com (G.D.Z.); m.j.m.cramer@umcutrecht.nl (M.-J.C.); a.j.teske-2@umcutrecht.nl (A.J.T.); l.n.broekhuizen-5@umcutrecht.nl (L.B.); vanderbilt@cardionetworks.org (I.v.d.B.); 10Department of Translational Medicine, University of Eastern Piedmont, Maggiore della Carità Hospital, 28100 Novara, Italy; 11Postgraduate School of Radiodiagnostics, Università degli Studi di Milano, 20122 Milan, Italy; serena.carriero@gmail.com; 12Department of Cardiology, Haga Teaching Hospital, 2545 GM The Hague, The Netherlands; 13School of Medicine and Surgery, University of Milano-Bicocca, 20126 Milan, Italy; g.muscogiuri@gmail.com (G.M.); sandro.sironi@unimib.it (S.S.); 14Department of Radiology, IRCCS Istituto Auxologico Italiano, San Luca Hospital, 20149 Milan, Italy; 15Department of Radiology, ASST Papa Giovanni XXIII Hospital, 24127 Bergamo, Italy; 16Department of Radiology, IRCCS Azienda Ospedaliero-Universitaria di Bologna, Via Albertoni 15, 40138 Bologna, Italy; luigi.lovato@aosp.bo.it

**Keywords:** chronic coronary syndrome, echocardiography, cardiac magnetic resonance, coronary computed tomography angiography, nuclear medicine, ischemia, CAD

## Abstract

Coronary artery disease (CAD) is one of the major causes of mortality and morbidity worldwide, with a high socioeconomic impact. Currently, various guidelines and recommendations have been published about chronic coronary syndromes (CCS). According to the recent European Society of Cardiology guidelines on chronic coronary syndrome, a multimodal imaging approach is strongly recommended in the evaluation of patients with suspected CAD. Today, in the current practice, non-invasive imaging methods can assess coronary anatomy through coronary computed tomography angiography (CCTA) and/or inducible myocardial ischemia through functional stress testing (stress echocardiography, cardiac magnetic resonance imaging, single photon emission computed tomography—SPECT, or positron emission tomography—PET). However, recent trials (ISCHEMIA and REVIVED) have cast doubt on the previous conception of the management of patients with CCS, and nowadays it is essential to understand the limitations and strengths of each imaging method and, specifically, when to choose a functional approach focused on the ischemia versus a coronary anatomy-based one. Finally, the concept of a pathophysiology-driven treatment of these patients emerged as an important goal of multimodal imaging, integrating ‘anatomical’ and ‘functional’ information. The present review aims to provide an overview of non-invasive imaging modalities for the comprehensive management of CCS patients.

## 1. A Multimodal Imaging Approach in Chronic Coronary Syndrome

Coronary artery disease (CAD) is a major cause of morbidity and mortality worldwide, and an accurate diagnostic assessment is pivotal for identifying patients that could potentially benefit from revascularization [1].

A multimodal non-invasive diagnostic approach for CAD detection includes anatomical (Coronary Computed Tomography Angiography, CCTA) and non-invasive functional imaging (stress echocardiography—SE, Cardiac Magnetic Resonance—CMR, nuclear imaging, stress CCTA, or CCTA derived Fractional Flow Reserve-FFR) [2,3].

While CCTA accurately depicts coronary anatomy, detects potential stenosis, and evaluates plaque features, other non-invasive functional imaging can demonstrate myocardial ischemia and the corresponding coronary territory. Integrating this complementary information is essential for the global risk assessment and the subsequential management of patients with suspected CAD [2].

The current European Society of Cardiology (ESC) guidelines recommend the use of either anatomical or non-invasive functional imaging as the initial test for diagnosing CAD after a global clinical risk assessment [4].

Which test to prescribe can be sometimes difficult to decide; typically CCTA is the preferred test in patients with a lower range of clinical likelihood of CAD, whereas the non-invasive functional tests for ischemia have better rule-in power and should be therefore preferred in those with higher clinical risk of coronary atherosclerosis [4]. The aim of this narrative review is to evaluate the potential strengths and novelty in the different modalities of cardiovascular imaging in the setting of ischemic heart disease and provide clinicians with practical advice for the diagnostic approach in patients with suspected CAD.

## 2. The Role of Echocardiography

Functional tests designed to identify suspected CAD, with or without cardiac imaging, traditionally serve three important clinical purposes in the field of cardiology:Diagnosing CADGuiding appropriate therapy (revascularization and/or medical intervention) in cases where CAD is confirmedAssessing the long-term outcomes and stratifying the risk for patients with CAD.

Stress echocardiography is commonly utilized as a non-invasive functional test during the diagnostic evaluation of CAD. It possesses the following characteristics:Cost-effectiveness and wide accessibilityAbsence of ionizing radiation (environmentally friendly)Ease of performance, potentially at the bedsideHigh diagnostic accuracy, particularly in terms of specificity for severe/obstructive CAD [1].

Stress echocardiography aims to detect myocardial ischemia by observing the transient changes in regional function that occur during stress. Specifically, the presence of coronary obstruction leads to reduced blood flow in the sub-endocardial region, resulting in decreased wall thickening and endocardial excursion in the ischemic areas. Stress echocardiography is performed using either exercise or pharmacologic stressors (dobutamine or vasodilator drugs) to induce myocardial ischemia. Furthermore, SE offers relevant information regarding coronary microvascular function, heart valves, or myocardium [5,6,7,8,9,10]. This imaging modality is limited in the case of poor acoustic windows. In that case, the application of ultrasound contrast media determines better the endocardial border delineation, allowing a precise assessment of wall thickening and excursion [11,12] (Figure 1).

Exercise, dobutamine, and vasodilators administered at adequately high doses have comparable effectiveness in inducing wall abnormalities in the presence of critical epicardial coronary stenosis [13]. However, in clinical practice, pharmacological SE is preferred over exercise SE [2,13]. This preference is primarily due to the potential physical demands associated with exercise SE, particularly in certain categories of patients, such as the elderly or those in whom exercise testing is not feasible [13]. Furthermore, exercise-induced hyperventilation and excessive chest wall motion can compromise the quality of SE examinations in some cases. However, exercise SE is recommended for active patients who have contraindications to dobutamine or vasodilators [2,13].

Despite the distinct pathophysiological mechanisms involved, dipyridamole and dobutamine tests demonstrate similar diagnostic accuracy when appropriately high doses and state-of-the-art protocols are employed [13]. In routine clinical practice, the choice of pharmacological agents is based on local expertise and specific contraindications, such as avoiding vasodilators in cases of severe asthma [2,13].

Vasodilator SE provides the opportunity to incorporate measurements of coronary flow velocity reserve in the left anterior descending coronary artery (CFVR-LAD). This measurement is an important parameter that enhances the specificity of SE evaluation [1,14,15] (Figure 2). However, it can be challenging to measure in certain cases and may also reflect impairment in the microvasculature, not solely epicardial stenosis [5,13].

Furthermore, SE can be implemented with speckle-tracking echocardiography (STE). The evaluation of regional wall motion abnormalities (WMA) is based only on myocardium inward motion; STE offers additional information regarding longitudinal myocardial shortening, has been shown to be superior to the visually assessed regional WMA, and is a useful tool for ischemia detection in patients with suspected chronic coronary syndrome [2,3].

In this setting, the Artificial Intelligence-calculated Left Ventricular Ejection Fraction and Global Longitudinal strain (GLS) are demonstrated to provide incremental sensitivity to detect CAD [16]. Artificial intelligence analysis pipelines efficiently delineate the endocardial surface without requiring advanced training for strain analysis. This reduces both variability and workload and ameliorates diagnostic performance [16].

Overall, SE demonstrates very high specificity compared to other functional tests for severe obstructive CAD detection (Figure 1 and Figure 2) and shows a high positive predictive value also in the low-risk population [17,18]. On the other hand, the sensitivity (and negative predictive value) of SE is low [19,20,21], and even lower if intermediate, or non-obstructive, CAD is concerned [18,22].

From a clinical perspective, if a SE is positive for ischemia and ≥2 segments of the left ventricle show reversible wall motion abnormalities, obstructive CAD is usually diagnosed. On the other hand, the diagnosis of less than severe CAD, which may also cause angina symptoms or cardiac events, is difficult to obtain with standard SE. The global (obstructive or not) CAD burden is clearly better assessed and quantified with CCTA, although a reduced CFVR-LAD during SE may improve the sensitivity for CAD with proven diagnostic benefits [23,24,25].

Stress-echocardiography has an established prognostic and risk stratification role, not only considering wall motion abnormalities, but also using CFVR-LAD measurement, and other ancillary variables, such as contrast myocardial perfusion [15,26,27].

Nevertheless, the question of whether SE offers comparable or superior long-term risk stratification compared to anatomical coronary data remains unknown due to the lack of direct comparative studies. Previous studies such as PROMISE and SCOT-HEART have compared coronary CCTA with a combination of various functional tests, but a direct head-to-head examination of SE against anatomical data is lacking [28,29].

In conclusion, SE continues to be a preferred non-invasive approach for patients with suspected CAD due to its wide availability. Furthermore, SE enables a comprehensive evaluation of cardiac function and the assessment of hemodynamic effects in patients with valvulopathies.

## 3. The Role of Cardiac Magnetic Resonance

Among noninvasive imaging modalities, CMR has been increasingly used in recent years due to its unique qualities in providing a complete assessment in patients with known or suspected CAD [30]. Indeed, CMR is able not only to evaluate the morphology, volume, and wall motion of the left ventricle (LV) but also allows precise tissue characterization. Furthermore, after gadolinium-based contrast medium (GBCM) injection, it enables the evaluation of stress-perfusion defects and the acquisition of late gadolinium enhancement (LGE) sequences for the detection of the extent of infarct scar [31,32,33].

In some cases, nuclear imaging may show false negative (balanced ischemia) or false positive results; in these circumstances, stress-CMR offers better sensibility and specificity in detecting functionally significant CAD [2].

In a few large randomized clinical trials, stress-CMR has been shown to be even superior in detecting ischemia compared to single photon emission computed tomography (SPECT) [34,35,36], and it resulted in a lower probability of unnecessary subsequent angiography [37,38]. The importance of stress-CMR is not only related to its high diagnostic accuracy but also to its ability to predict the patient’s prognosis.

In fact, numerous studies have demonstrated that pathological stress-CMR is associated with a higher risk of cardiac death and adverse events during long-term follow-up [39,40,41,42].

In patients with prior revascularization, a stress-CMR reduces the need for further diagnostic imaging techniques, subsequent coronary angiography, and revascularizations without impairing patients’ prognosis [30].

Stress-CMR can be performed after the injection of a vasodilator substance (i.e., adenosine, regadenoson, or dipyridamole) which triggers a “coronary steal effect”: In normal myocardium, coronary microcirculation dilates during exercise ensuring adequate tissue perfusion, while in the presence of significantly stenotic coronary arteries, the distal microcirculation is almost maximally dilated even in resting conditions. During or after vasodilator injection, first-pass transit of GBCM is observed through the left ventricular myocardium with a typical wavefront from the subendocardial to the subepicardial region. Typically, 3 short-axis slices are acquired for each heartbeat, and the whole first-pass perfusion scan is performed during one breath-hold [43,44]. Currently, perfusion deficits are commonly assessed visually by expert physicians (Figure 3). However, with the support of artificial intelligence, semi-quantitative and quantitative methods have been developed and may offer valuable assistance in detecting perfusion defects.

Phase contrast CMR at the coronary sinus allows a valuable estimation of the global left ventricle (LV) myocardial blood flow. This imaging technique has been validated in small studies using positron emission tomography (PET). Recent studies demonstrated a prognostic role of CMR-derived Coronary Flow Reserve (CFR) in CAD patients [45,46].

Stress-CMR offers a global assessment of myocardial ischemia and myocardial viability in a single examination (Figure 3) [47]. The dynamic accumulation of GBCM in different areas of the myocardium reflects the pathophysiological process of the ischemic wavefront and has a typically subendocardial or transmural distribution in ischemic heart disease. In addition, te GBCM is distributed in the extracellular volume of the myocardium, which appears particularly represented in the case of scar tissue and therefore is accumulated in infarct tissue. Infarct scars that do not exceed 25% of myocardial wall thickness are most likely to achieve functional recovery after revascularization, while segments with LGE extension of more than 50% are unlikely to recover [48]. It is also known that infarct size evaluated by LGE-CMR is by far the best predictor of mortality and significant cardiac events [49]. Moreover, LGE location, burden, and the presence of transmural necrosis have been correlated to response to cardiac resynchronization therapy and arrhythmic risk [50].

In cases where vasodilators are contraindicated, the assessment of myocardial ischemia can be conducted through dobutamine infusion. Dobutamine, a positive inotropic and chronotropic agent, raises myocardial oxygen demand, potentially inducing ischemia and resulting in left ventricular wall motion abnormalities in individuals with significant CAD. The protocol for this assessment is similar to that of SE, involving the administration of escalating doses of dobutamine until the target heart rate, equivalent to 85% of the maximal predicted heart rate, is achieved [2]. Dobutamine stress-CMR is useful in patients with severe renal disease or in cases where vasodilators or GBCM are contraindicated [47]. If typically excluded from large prognostic studies, recently CMR has been found to have a sensibility and specificity in detecting perfusion defects also in patients with previous ischemic heart disease, atrial fibrillation, and with MR-compatible implantable devices [51,52,53,54]. As CMR is increasingly used and available for the study of ischemic heart disease, new technologies are implementing this method to expand its use. One of the most promising methods is the blood oxygen level–dependent (BOLD) CMR, which uses the paramagnetic properties of deoxyhemoglobin as an endogenous contrast agent with increased deoxyhemoglobin content leading to a signal reduction on T2* or T2-weighted images. Thus, BOLD CMR directly reflects myocardial oxygenation status [55,56]. Finally, possible future solutions are to combine functional 3D-CMR perfusion data with anatomical 3D-CMR coronary angiography images performed within a single exam and hybrid imaging, such as PET/CMR, which combined strengths of each imaging modalities, in particular, the accuracy of quantitative myocardial blood flow with PET with the high spatial resolution of the CMR [57,58].

## 4. The Role of Nuclear Medicine (SPECT/PET)

The use of single photon emission computed tomography (SPECT) is currently an established approach in the initial evaluation of patients with suspected CAD thanks to wide availability, standardized protocols, and the extensive data established for diagnostic accuracy. Ischemia can be provoked by exercise or pharmacological stressors (dobutamine) that increase myocardial work and oxygen demand, as in other stress imaging protocols [2]. In case of left bundle branch block or ventricular paced rhythms, vasodilators (i.e., adenosine, regadenoson, or dipyridamole) should be preferred to identify heterogeneity in myocardial perfusion as well as in patients who are not able to achieve ≥85% of maximal age-predicted heart rate during exercise [59]. The mechanisms of action of the vasodilator substances and dobutamine are the same as already described in the CMR section.

Single-photon emission computed tomography implicates the intravenous administration of gamma-emitting radiotracers, which are accumulated by cardiomyocytes in proportion to myocardial blood flow. This uptake occurs during periods of rest as well as during physical or pharmacological stress. The radionuclide agent is typically injected at the peak of exercise or during maximum vasodilation. During stress, a decrease in regional tracer uptake indicates relative myocardial hypoperfusion, whereas reduced uptake both during stress and at rest suggests the presence of a myocardial scar [60]. Two radiopharmaceuticals labeled with technetium-99m (99mTc) (sestamibi and tetrofosmin) and thallium-201 (201Tl) chloride are currently commercially available [59,61].

The particular pharmacokinetic characteristic of 201Tl is the prolonged retention within the cardiomyocytes allowing the evaluation of the coronary reserve in a single administration immediately after the provocation test. On the contrary, the perfusion study should be performed in two sessions (stress and rest test, usually 24 h apart) using the technetium-based tracers, with more favorable dosimetry and improved quality of gated images thanks to their shorter half-life [59].

Currently, myocardial perfusion studies are acquired in a gated mode using an ECG trigger and separate computed tomography (CT) scans are performed for attenuation correction (available for SPECT/CT hybrid machines). This hybrid modality offers the evaluation of the ventricular volumes and function (including the regional contractility of the LV) with an improvement in the diagnostic accuracy of perfusion imaging [62].

Unfortunately, there are some limitations that need to be mentioned. First, SPECT can overlook cases of balanced ischemia [63]. Furthermore, SPECT images suffer from several artifacts that mimic perfusion defects, as in cases of left bundle branch block, which can cause an apparent defect in the septal wall, of the breast and diaphragm interposition for the anterior wall or inferior wall, respectively. Subdiaphragmatic hepatobiliary excretion of technetium-labeled agents may mimic an increased activity in the inferior wall [62]. Finally, it is a global long examination that suffers from higher radiation exposure, although newer technologies such as cardiac-specific solid-state detector cameras work at lower radiation doses [64]. In the future, the development of new SPECT imaging systems with greater sensitivity, compact design, and new reconstruction algorithms associated with new Tc-99m-labeled deoxyglucose will improve image quality and resolution, expanding the role of SPECT in ischemic heart disease diagnosis [65].

Positron emission tomography (PET) is another radionuclide imaging technique widely used in the study of ischemic heart disease and other cardiovascular pathologies [2,66]. This technique assesses both perfusion and metabolism function thanks to the use of tracers with different pharmacokinetic properties.

Myocyte metabolism shifts to glucose from fatty acids during ischemia. Thus, the lower uptake of a glucose analogue tracer (18F-Fluorodeoxyglucose—FDG) reflects the presence of myocardial ischemia. To evaluate regional perfusion, a tracer that remains in the vascular space and illustrates the distribution of myocardial blood flow (such as nitrogen 13-ammonia or rubidium-82) can be utilized. Consequently, the presence of enhanced FDG uptake in regions with reduced blood flow, known as a “mismatch,” signifies hibernated myocardium. Conversely, when both metabolism and flow show a consistent decrease, termed a “match,” it is believed to indicate necrotic myocardium. The presence of regional dysfunction alongside normal perfusion is an indication of stunning [2].

The advantage of PET imaging is that the myocardial perfusion can be measured directly and quantified in all left ventricular myocardium. This allows the identification of ischemia even in the case of multivessel disease as well as the assessment of microvascular dysfunction [67].

PET with myocardial perfusion imaging (PET-MPI) has some other advantages over SPECT such as lower radiation dose, better image quality, and greater diagnostic accuracy [68]. In fact, some myocardial segments that appear severely hypoperfused on SPECT demonstrate FDG uptake. However, direct comparisons of PET and SPECT in broad groups of patients with a wide range of LV systolic functions are lacking. In conclusion, PET scanners are associated with higher costs, and the availability of PET tracers is more limited compared to SPECT tracers, thereby restricting their utilization. However, the future advancement and introduction of novel PET tracers have the potential to address these limitations [68].

## 5. The Role of Coronary Computed Tomography Angiography

Coronary CT Angiography (CCTA) is the preferred imaging technique in symptomatic patients with a low-intermediate pre-test probability of CAD [4,69]. The current new-generation CT scanners enable high image quality with reduced contrast volume and radiation dosage, providing high diagnostic accuracy in the detection of CAD, even in the case of patients with high and/or irregular heart rates [20,70]. Integrating CCTA in the diagnostic algorithm of patients with stable chest pain was shown to be associated with a significant reduction in cardiovascular death and non-fatal myocardial infarction, due to proper diagnosis and tailoring of the treatment strategy [71,72].

Coronary CT Angiography allows risk stratification of patients with CAD based on the severity of coronary stenoses but also evaluation of the plaque characteristics (Figure 4). Importantly, baseline plaque burden is associated with the risk of major cardiovascular events independently of the presence of obstructive lesions [73,74,75] and predicts the progression to obstructive CAD [76]. High-risk plaque features (i.e., positive remodeling, low attenuated plaques, napkin-ring sign, spotty calcification) are strong predictors of future MI and focused the physician on secondary medical treatments [77,78,79].

The specificity of CCTA in CAD is boosted with the integration of anatomical information with functional hemodynamic assessment (Figure 4), which includes (i) CT-derived fractional flow reserve (FFRCT) and (ii) CT perfusion (CTP) [80]. The use of FFRCT, which is based on computational fluid dynamics, showed the highest diagnostic performance for vessel-specific ischemia compared with SPECT and PET [81]. Moreover, in the case of FFRCT < 0.80, stable symptomatic patients diagnosed with CAD at CCTA have been shown to experience significantly lower cardiovascular death or myocardial infarction [82]. Thus, the diagnostic strategy based on CCTA integrated with FFRCT proves to be accurate and cost-effective, reducing the number of unnecessary invasive coronary angiography [28]. In the PERFECTION Trial, the addition of both FFRCT or stress-CTP to CCTA improved its diagnostic accuracy and positive predictive value in the evaluation of the functional relevance of CAD [83]. Compared with computational fluid dynamics (CFD) analytics, CTP has limited application in clinical practice and data regarding prognostic implications are lacking. However, the combined approach CCTA plus CTP is characterized by excellent specificity for the detection of hemodynamically significant CAD [84,85]. Moreover, it is comparable to invasive coronary angiography plus SPECT in predicting major cardiovascular events [86].

## 6. Perivascular Adipose Tissue (PVAT)

Perivascular adipose tissue (PVAT) is a fat depot with a paracrine function that produces a wide range of biologically active molecules, which may profoundly influence the vasculature itself [87,88].

PVAT attenuation, measured with CT imaging is a non-morphological marker of inflammation that was applied to the left atrium, carotid artery, and aorta [89,90,91]. The role of coronary inflammation in the pathogenesis of atherosclerotic plaque instability is now well-established [92,93].

Inflamed vessels release cytokines that prevent lipid accumulation in PVAT preadipocytes and play a key role in the progression of vascular atherosclerosis. Therefore, in the presence of vascular inflammation adipogenesis is inhibited in favor of lipolysis, and water content increases in the adipose cells. This process shifts overall PVAT attenuation on CT (measured in Hounsfield unit—HU) due to oedema and it is a useful biomarker for the in-vivo assessment of coronary inflammation [94]. Oikonomou et al. demonstrated the independent prognostic value of the PCAT attenuation around the proximal right coronary artery and left anterior descending artery at long-term follow-up. The cut-off for the PVAT attenuation of –70.1 HU has been shown to be correlated with relevant worse outcomes [95]. In addition, PVAT attenuation is greater in patients affected by myocardial infarction with non-obstructive coronary arteries (MINOCA) and Tako-Tsubo syndrome than in controls, underlining the role of inflammation in the genesis of vascular pathologies [96,97,98,99].

The diagnosis of ischemia with no obstructive coronary arteries (INOCA) is currently challenging and is based on the demonstration of coronary microcirculatory dysfunction in the absence of CAD [100,101]. Recently it has been demonstrated that in patients with INOCA, there is an association between the reduction in PVTA attenuation and CFVR-LAD [102].

## 7. CCTA: Emerging Techniques and Future Perspectives

Coronary CT Angiography is expected to improve the diagnosis and management of patients with CCS, thanks to the evolution of CFD analytics, advances in hardware, and quantitative CT analysis. Besides vessel-related ischemia, CFD allows the evaluation of hemodynamic forces acting on coronary plaques, which could be related to plaque rupture and thus to the risk of developing acute coronary syndromes (ACS). The EMERALD trial has demonstrated that delta FFR_CT_ across the lesion is the strongest predictor of acute coronary syndrome, with a small incremental prognostic value provided by axial plaque and wall shear stress over plaque morphology alone [103]. A novel tool derived from FFR_CT_, the FFR_CT_ Planner, allows virtual stenting of coronary stenosis and real-time prediction of post-angioplasty FFR, which could affect the patient selection and procedural planning [104]. Plaque characterization will be consistently improved by the advent of true cardiac-capable photon counting detectors, which have far superior spatial resolution with reduced radiation dose [105]. The derived huge amount of CT data available could be quantitatively analyzed via artificial intelligence, machine learning, and radiomics to refine risk prediction models of clinical outcomes [106,107]. All these innovations are paving the way for the shift of CCTA in the field of interventional cardiology, particularly for preprocedural planning, intraprocedural guidance, and tailoring of the revascularization strategy [108,109,110].

## 8. Functional Versus Anatomical Imaging

Non-invasive multimodality imaging in chronic coronary syndrome offers a comprehensive anatomic and functional assessment through direct visualization of coronary arteries and a precise quantification of myocardial ischemia or viability. Furthermore, modern techniques can now accurately evaluate functionally coronary microcirculation and diagnose coronary microvascular dysfunction (CMD) [2,101].

According to the 2019 ESC Guidelines for Coronary Chronic syndromes revascularization is indicated whenever significant stenosis or inducible ischemia are demonstrated at imaging techniques [4].

Both anatomic and functional imaging techniques have strengths and limitations: ESC guidelines recommend CCTA as the initial diagnostic test in patients with a low-to-intermediate clinical probability of CAD due to its high negative predictive value. On the other hand, in patients with an intermediate-high clinical likelihood of CAD a functional ischemia test is preferred as the basis for subsequent coronary angiography [111]. This strong indication is based on previously published, non-randomized studies which showed how patients with a significant (>10%) amount of ischemic myocardium during stress would benefit from revascularization [111,112]. Nevertheless, it must be pointed out that these observations were not confirmed by more recent studies; notably, the ISCHEMIA trial demonstrated that in patients affected by stable coronary artery disease and moderate-to-severe ischemia, an initial revascularization strategy, either surgical or percutaneous, did not reduce cardiovascular events or death from any causes during follow up compared to an initial conservative approach. In addition, unprotected left main disease (≥50%) was excluded in the study population by an initial CCTA [113].

On the other hand, in this trial, CCTA showed superior diagnostic performance compared to stress imaging as it was able to rule out significant CAD in 20% of patients with moderate-to-severe ischemia [111,113].

Furthermore, the paradigm of myocardial hibernation has been recently challenged by the REVIVED trial, which demonstrated that percutaneous revascularization did not result in a lower incidence of all-cause death or hospitalization for heart failure in patients with severe ischemic left ventricular dysfunction (Left Ventricular Ejection Fraction < 35%) and demonstrable myocardial viability at functional imaging, as compared to optimal medical therapy alone [114].

Notwithstanding the growing relevance of anatomical imaging, some aspects should be considered. In these trials, ischemia is predominantly demonstrated through low specificity and sensitivity techniques such as treadmill tests: it could be argued that ischemia could significantly affect patients’ prognosis when assessed by more sophisticated functional tests, as suggested in patients with angina and non-obstructive coronary arteries [115]. Recently, evidence showed that stress CMR had incremental prognostic value in symptomatic patients with obstructive CAD of unknown significance on CCTA [116]. Furthermore, due to its exceptional specificity, functional imaging plays a crucial role in accurately diagnosing suspected CAD patients with a moderate-to-high clinical likelihood. [4].

Functional non-invasive imaging is essential in some subcategories. In fact, patients with suspected CAD and valvular diseases should undergo stress echocardiography for the assessment of ischemia and the effective hemodynamic impact of valvulopathy [117]. On the other hand, stress CMR offers an accurate tissue characterization which is pivotal to identifying potential differential diagnoses such as myocarditis or cardiomyopathies [118,119]. Furthermore, functional imaging represents a valuable alternative in case of inconclusive anatomical imaging or contraindications to CCTA (renal failure, high-rate atrial fibrillation) [2]. Finally, non-invasive anatomical and functional imaging techniques are essential and synergistic in the diagnostic workup of chronic coronary syndrome without obstructive coronary arteries, as the former excludes CAD and the latter demonstrates ischemia and microvascular dysfunction, both necessary criteria for the diagnosis of microvascular angina [100].

Recent guidelines still recommend both functional and anatomical imaging in patients with suspected CAD based on their clinical likelihood. Both pathways offer complementary information for the global clinical diagnosis and subsequent management of suspected CAD [4,120].

## 9. Conclusions and Future Directions

In conclusion, although the most recent evidence hints towards anatomical imaging, it has to be outlined that a global assessment of the patient is often necessary and that functional imaging offers valuable information pivotal to the further therapeutic pathway. Non-invasive imaging methods for the diagnosis of CAD have distinct characteristics; the patient’s cardiovascular risk assessment and pre-test probability should guide the choice of the best method.

Cardiologists and radiologists should therefore be aware of the strengths and weaknesses of these imaging techniques in order to choose the diagnostic pathway that tailors properly to the specific patient (Figure 5). Future larger studies are needed to evaluate the proper indication of anatomical or functional imaging to guide the best management of CCS patients.

## Figures and Tables

**Figure 1 diagnostics-13-02083-f001:**
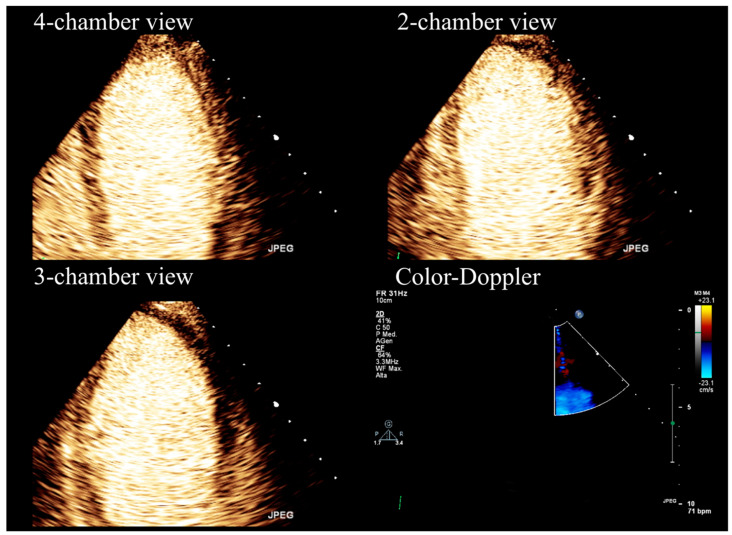
Stress echocardiography performed using high-dose (0.84 mg/kg/6 min) dipyridamole and contrast-enhanced imaging.

**Figure 2 diagnostics-13-02083-f002:**
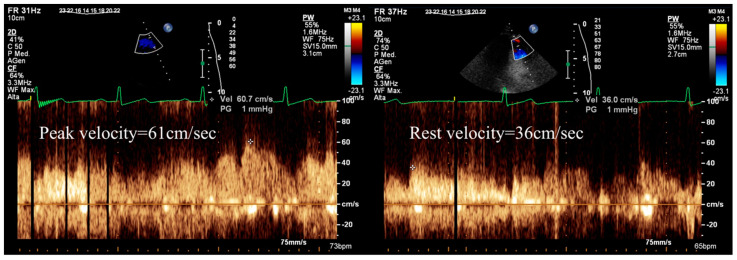
Calculation of Doppler CFVR-LAD during stress echocardiography, demonstrating reduced stress (left) to rest (right) peak diastolic velocity ratio (*61/36 = 1.7; normal value > 2.0), suggesting significant, probably obstructive, CAD in the left anterior descending artery.

**Figure 3 diagnostics-13-02083-f003:**
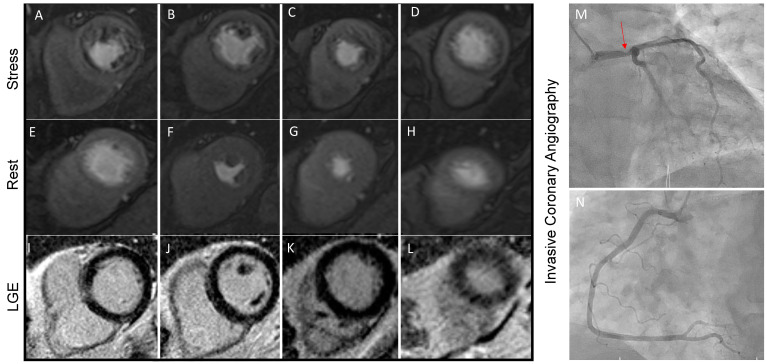
Stress perfusion CMR performed on a 63-year-old man with multiple cardiovascular risk factors and symptomatic for shortness of breath during moderate physical efforts. Panels (**A**–**D**) show the presence of diffuse hypoperfusion during regadenoson infusion that is not present during the rest phase (Panel (**E**–**H**). Since no scar is detected in LGE sequences (Panel (**I**–**L**)), the stress perfusion CMR concluded for large perfusion defects, which was confirmed by invasive coronary angiography. A severe stenosis of the Left Main Steam (Panel (**M**), red arrowed) was found. Panel (**N**) is showing right coronary artery, which has no critical stenosis.

**Figure 4 diagnostics-13-02083-f004:**
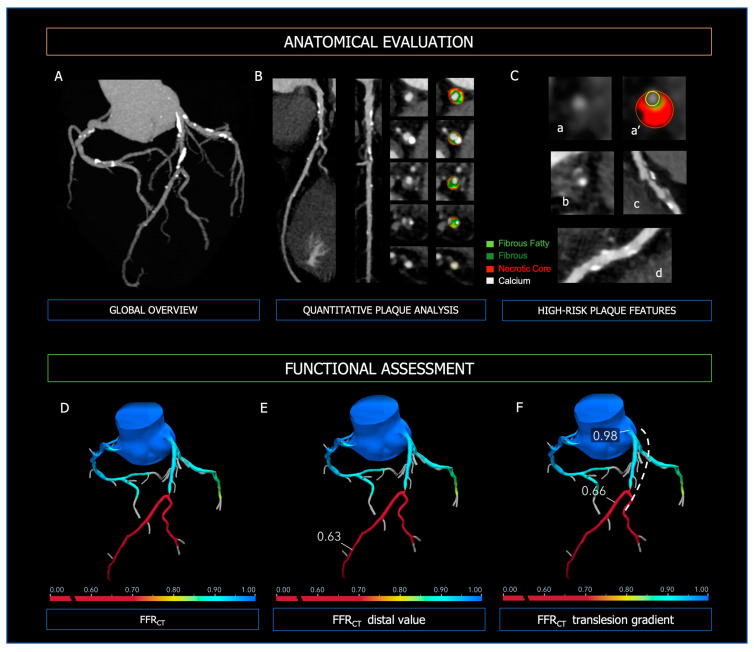
Anatomical evaluation and functional assessment of coronary plaques derived from CCTA. Panel (**A**)—3-vessel overview of the coronary tree for evaluation of calcium extent and distribution; Panel (**B**)—curved MPR, straight MPR, and cross-sections for quantitative plaque analysis; Panel (**C**)—high-risk plaque features; (**a**,**a’**): low attenuated plaque; (**b**): napkin-ring sign; (**c**): positive remodeling; (**d**): spotty calcifications. In the upper Panels, the figure legend refers to the colors used for quantitative plaque analysis; light green: fibrous fatty volume; dark green: fibrous volume; red: necrotic core volume; white: calcific volume. Panel (**D**): FFR-CT 3D model of the coronary tree, with distal FFR-CT (Panel (**E**)) and trans-lesion gradient (Panel (**F**), dashed line). At the bottom of Panels (**D**–**F**), the colorimetric scale of FFR-CT used in the 3D models is shown.

**Figure 5 diagnostics-13-02083-f005:**
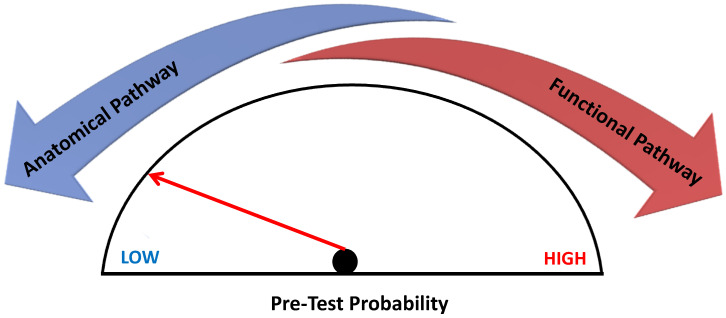
Non-invasive imaging in Chronic Coronary Syndrome. Non-invasive imaging in chronic coronary syndrome according to pre-test probability. The anatomical diagnostic pathway is indicated in patients with low to intermediate pre-test probability of CAD according to current guidelines, whereas in the case of intermediate-high probability, it is better to choose a functional imaging test.

## Data Availability

Not applicable.

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
