# Peer review of "The Role of Non-Invasive Multimodality Imaging in Chronic Coronary Syndrome: Anatomical and Functional Pathways"

_diagnostics, 2023, doi:10.3390/diagnostics13122083_

Round 1

Reviewer 1 Report

Multimodality imaging, including echocardiography, cardiac computed tomography, cardiac magnetic resonance and nuclear cardiology, has fundamentally advanced the understanding and treatment of cardiovascular disease the last two decades. In the 21st century, it is essential to use and integrate these different modalities to the best advantage of patients.

In this state-of-the-art review, Authors summarize the evolution of multimodality cardiac imaging in chronic coronary syndrome in the 21st century, highlighting the opportunities for future innovations.

The paper is well written, well-organized, well-illustrated and interesting to read; references in the bibliography are rich and updated.

Sometimes it is difficult to read due to some abbreviations, used without a prior definition.

Just few comments

Page count is not correct: after “5 of 20” we have two pages without number and then “2 of 20” again, so that the last page of the manuscript is “14 of 20”, instead of “20 of 20”. I’ll use, therefore, just the line count to indicate my comments.

-       Line 245: incorrect grammar ( ; instead of , ).

-       Line 256: in order to read more easily the paper, it would be valuable to clearly explain some abbreviations like PET, similarly to SPECT some lines before.

-  Line 217: in this paragraph, Authors should include the latest advanced technology in hybrid imaging, that is PET/MRI, which has the potential to combine the individual strengths of PET and MRI in the assessment of MPI, in particular the accuracy of quantitative MBF with PET, the high spatial resolution of MRI, and the use of cine MRI to assess for regional wall motion abnormalities.

-       Line 298: focused instead of focuse.

-       Line 319: abbreviation “CFD” is used without a prior definition.

-       Line 349: abbreviation “CFVR-LAD” is defined in the abbreviation list, but the paragraph should probably be self-contained.

Author Response

We thank the Reviewer for these precious comments.

In the updated version of the manuscript, we corrected and modified the text accordingly to the Reviewer’s suggestion.

We hope that all issues have been resolved and we believe that the document has been consistently improved.

----------------------------------------------------------------------

Reviewer 2 Report

Dear Editor,

The article "The role of multimodality imaging in chronic coronary syndrome: anatomical and functional pathways" is an interesting and timely topic where the authors describe the state of the art regarding the value and limitations of non-invasive methods that can assist in the early diagnosis of coronary artery disease, allowing for more rational and effective treatment. The text is easy to read, and the English is flawless.

Author Response

We greatly appreciate the reviewer for this positive feedback.

----------------------------------------------------------------------

Reviewer 3 Report

CAD, in terms of both diagnosis and treatment, is a topic of major interest for many categories of doctors who participate in the care of patients with cardiovascular disease. The topic addressed is both current and of wide interest. The systematic and practical approach is one of the strengths of the manuscript.

Given that a review is comprehensive and detailed, some improvements are recommended:

Section 2: The role of echocardiography

It would be interesting to comment:

on the benefits and limits of SE - effort vs pharmacological;

on the benefits and limits of pharmacological SE - dobutamine vs vasodilator.

Please add a comment on the limits of CFVR-LAD.

A prioritization among SE techniques would be useful (the practical approach).

Considering that other modern techniques for evaluating coronary arteries exist, it would be useful to introduce a paragraph that comments on intravascular ultrasound (IVUS) and optical coherence tomography (OCT).

Thank you!

Author Response

Reviewer 3

CAD, in terms of both diagnosis and treatment, is a topic of major interest for many categories of doctors who participate in the care of patients with cardiovascular disease. The topic addressed is both current and of wide interest. The systematic and practical approach is one of the strengths of the manuscript.

Given that a review is comprehensive and detailed, some improvements are recommended:

Section 2: The role of echocardiography

It would be interesting to comment:

on the benefits and limits of SE - effort vs pharmacological;

on the benefits and limits of pharmacological SE - dobutamine vs vasodilator.

Please add a comment on the limits of CFVR-LAD.

A prioritization among SE techniques would be useful (the practical approach).

We updated the manuscript accordingly to the Reviewer’s suggestion, as follows:

“Exercise, dobutamine, and vasodilators administered at adequately high doses have comparable effectiveness in inducing wall abnormalities in the presence of critical epicardial coronary stenosis 13. However, in clinical practice, pharmacological SE is preferred over exercise SE 2,13. This preference is primarily due to the potential physical demands associated with exercise SE, particularly in certain categories of patients, such as the elderly or those in whom exercise testing is not feasible 13. Furthermore, exercise-induced hyperventilation and excessive chest wall motion can compromise the quality of SE examinations in some cases. However, exercise SE is recommended for active patients who have contraindications to dobutamine or vasodilators. 2,13

Despite the distinct pathophysiological mechanisms involved, dipyridamole and dobutamine tests demonstrate similar diagnostic accuracy when appropriately high doses and state-of-the-art protocols are employed.13 In routine clinical practice, the choice of pharmacological agents is based on local expertise and specific contraindications, such as avoiding vasodilators in cases of severe asthma. 2,13

Vasodilator SE provides the opportunity to incorporate measurements of coronary flow velocity reserve in the left anterior descending coronary artery (CFVR-LAD). This measurement is an important parameter that enhances the specificity of SE evaluation 1,15 (Figure 2). However, it can be challenging to measure in certain cases and may also reflect impairment in the microvasculature, not solely epicardial stenosis 5,13.”

We believe that the manuscript has improved considerably thanks to these suggestions.

Considering that other modern techniques for evaluating coronary arteries exist, it would be useful to introduce a paragraph that comments on intravascular ultrasound (IVUS) and optical coherence tomography (OCT).

We thank the Reviewer for this suggestion.

While we acknowledge the reviewer's viewpoint on the significance of invasive intracoronary imaging methods such as IVUS and OCT in CAD assessment, we believe that including a dedicated paragraph on this topic exceeds the scope of our review.

Therefore, we have made a revision to the title of our manuscript as follows:

“The role of non-invasive multimodality imaging in chronic coronary syndrome: anatomical and functional pathways”

Thank you